# Remapping in a recurrent neural network model of navigation and context inference

Isabel IC Low[1]*, Lisa M Giocomo[2], Alex H Williams[3,4]*

[1]Zuckerman Mind Brain Behavior Institute, Columbia University, New York, United States; [2]Department of Neurobiology, Stanford University, Stanford, United States; [3]Center for Computational Neuroscience, Flatiron Institute, New York, United States; [4]Center for Neural Science, New York University, New York, United States

**Abstract** Neurons in navigational brain regions provide information about position, orientation, and speed relative to environmental landmarks. These cells also change their firing patterns ('remap') in response to changing contextual factors such as environmental cues, task conditions, and behavioral states, which influence neural activity throughout the brain. How can navigational circuits preserve their local computations while responding to global context changes? To investigate this question, we trained recurrent neural network models to track position in simple environments while at the same time reporting transiently-cued context changes. We show that these combined task constraints (navigation and context inference) produce activity patterns that are qualitatively similar to population-wide remapping in the entorhinal cortex, a navigational brain region. Furthermore, the models identify a solution that generalizes to more complex navigation and inference tasks. We thus provide a simple, general, and experimentally-grounded model of remapping as one neural circuit performing both navigation and context inference.

## eLife assessment

This **important** work provides evidence that artificial recurrent neural networks can be used to investigate neural mechanisms underlying reversible remapping of spatial representations. Authors perform **convincing** state of the art analyses showing how population activity preserves the encoding of spatial position despite remappings due to the tracking of an internal variable. This paper will be of interest to neuroscientists studying contextual computations, neural representation of space and links between artificial neural networks and the brain.

## Introduction

Neural circuit computations throughout the brain, from the primary sensory cortex (***Bennett et al., 2013***; ***Niell and Stryker, 2010***; ***Vinck et al., 2015***; ***Zhou et al., 2014***) to higher cognitive areas, (***Boccara et al., 2019***; ***Butler et al., 2019***; ***Hardcastle et al., 2017b***; ***Hulse et al., 2017***; ***Pettit et al., 2022***) are shaped by combinations of internal and external factors. Internal state changes, such as shifts in attention (***Fenton et al., 2010***; ***Kentros et al., 2004***; ***Pettit et al., 2022***), thirst (***Allen et al., 2019***), arousal (***Stringer et al., 2019***), and impulsivity (***Cowley et al., 2020***), can profoundly alter neural activity across multiple brain areas. This raises a question: how can individual brain regions with specialized functions integrate global state changes without compromising their local processing dynamics?

*For correspondence:
il2419@columbia.edu (IICL);
alex.h.williams@nyu.edu (AHW)

**Competing interest:** The authors declare that no competing interests exist.

For example, neurons in the medial entorhinal cortex typically represent one or more features such as spatial position, heading direction, and environmental landmarks and are therefore thought to support navigation (*Diehl et al., 2017*; *Gil et al., 2018*; *Hafting et al., 2005*; *Hardcastle et al., 2017a*; *Høydal et al., 2019*; *Moser et al., 2014*; *Sargolini et al., 2006*; *Solstad et al., 2008*). At the same time, these neurons change their firing rates and shift their spatial firing positions—or 'remap'—under a variety of circumstances, even when navigational cues remain stable (*Bant et al., 2020*; *Boccara et al., 2019*; *Butler et al., 2019*; *Campbell et al., 2021*; *Campbell et al., 2018*; *Hardcastle et al., 2017b*; *Low et al., 2021*). It is difficult to pinpoint the reason for these *spontaneous remapping events*—i.e., remapping not driven by changes in navigational features. Theoretical models of this phenomenon propose that remapping occurs because these cells are responding to global contextual cues (like arousal or attention) in order to decorrelate related experiences with distinct contextual relevance (*Colgin et al., 2008*; *Sanders et al., 2020*). This process could enable animals to form distinct memories or choose appropriate actions for a given set of circumstances. However, these normative models (i.e. theories for *why* remapping occurs; *Levenstein et al., 2020*) do not address how a biological system might implement this strategy.

To bridge the gap between existing theoretical models and biological observations of remapping in the entorhinal cortex, we sought to establish a minimal set of task constraints that could reproduce the essential dynamics of remapping in a computational model. Specifically, we tested the normative hypothesis that remapping occurs when a population of neurons must maintain its local navigational processing, while at the same time responding to global latent state changes (e.g. changes in behavioral state, task conditions, etc.; see *Sanders et al., 2020*). We trained recurrent neural network models (RNNs) to maintain an estimate of position in a simple environment, while at the same time reporting a changing, transiently-cued latent state variable. In isolation, neither of these tasks is novel to the RNN literature—e.g., *Cueva and Wei, 2018* trained RNNs to path integrate in complex environments while *Sussillo and Barak, 2013* trained RNNs on a '1-bit flip-flop' memory task akin to our latent state inference task. Here, we combine these two tasks to ask how a network would solve them simultaneously and to probe how this combination of tasks relates to remapping in navigational circuits.

We found that RNNs trained to navigate while inferring latent state changes exhibited network-wide activity patterns that were strikingly similar to those found in the brain (*Low et al., 2021*), suggesting a possible function for spontaneous remapping in the entorhinal cortex and other navigational brain areas. These activity patterns comprise a geometrically simple solution to the task of combining navigation with latent state inference. The RNN geometry and algorithmic principles readily generalized from a simple task to more complex settings. Furthermore, we performed a new analysis of experimental data published by *Low et al., 2021* and found a similar geometric structure in neural activity from a subset of sessions with more than two stable spatial maps. Overall, these results provide an interpretable and experimentally grounded account of how a single neural population might flexibly represent global brain state changes (corresponding here to remapping) and localized circuit computations (corresponding here to navigation) in orthogonal subspaces (*Kaufman et al., 2014*; *Rule et al., 2020*).

## Results

### A recurrent neural network model of 1D navigation and context inference remaps between aligned ring manifolds

To investigate a putative functional role for spontaneous remapping in an unchanging environment, we developed a task that requires simultaneous latent state inference and navigation in a single neural circuit. To ground our model in experimental data, we designed our task to reflect the basic structure of a recent study (*Low et al., 2021*; *Figure 1A–D*). In this study, Low et al. demonstrated that remapping in the medial entorhinal cortex simultaneously recruited large populations of neurons across the entorhinal cortical circuit. Remapping comprised discrete transitions between aligned neural activity manifolds, which each represented a distinct map of an unchanging, virtual reality 1D environment (*Figure 1C–D*). Remapping was not aligned to particular track positions, rewards, or landmarks. Instead, remapping correlated with transient decreases in running speed (*Figure 1B*), which could correspond to discrete changes in a latent state (such as shifts in arousal, task engagement, or other

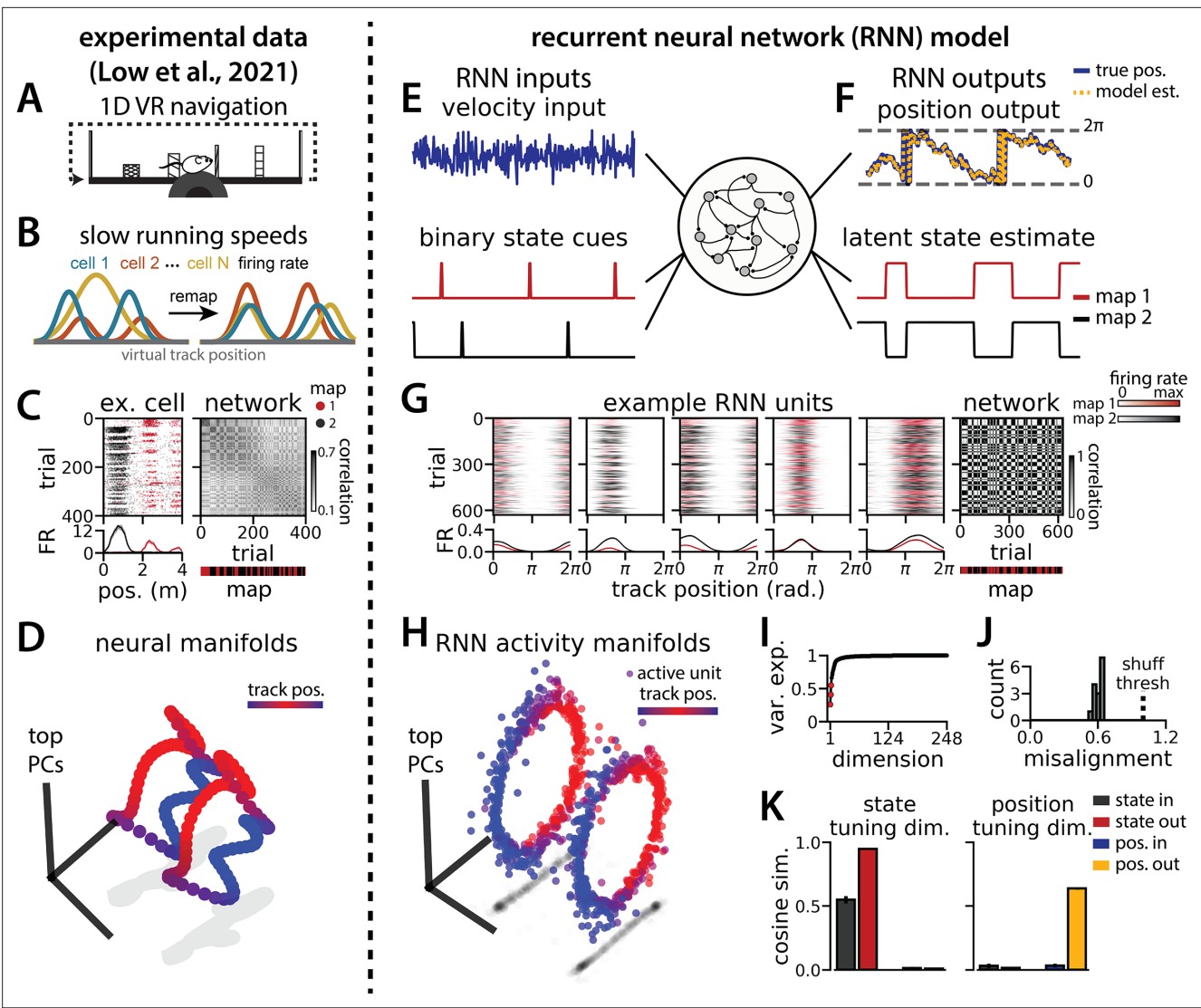

**Figure 1.** Recurrent neural network (RNN) models and biological neural circuits remap between aligned spatial maps of a single 1D environment. (**A–D**) are modified from *Low et al., 2021*. (**A**) Schematized task. Mice navigated virtual 1D circular-linear tracks with unchanging environmental cues and task conditions. Neuropixels recording probes were inserted during navigation. (**B**) Schematic: slower running speeds correlated with the remapping of neural firing patterns. (C, left) An example medial entorhinal cortex neuron switches between two maps of the same track (top, spikes by trial and track position; bottom, average firing rate by position across trials from each map; red, map 1; black, map 2). (C, right/top) Correlation between the spatial firing patterns of all co-recorded neurons for each pair of trials in the same example session (dark gray, high correlation; light gray, low correlation). The population-wide activity is alternating between two stable maps across blocks of trials. (C, right/bottom) K-means clustering of spatial firing patterns results in a map assignment for each trial. (**D**) Principal Components Analysis (PCA) projection of the manifolds associated with the two maps (color bar indicates track position). (**E**) RNN models were trained on a simultaneous 1D navigation (velocity signal, top) and latent state inference (transient, binary latent state signal, bottom) task. (**F**) Example showing high prediction performance for the position (top) and latent state (bottom). (**G**) As in (**C**), but for RNN units and network activity. Map is the predominant latent state on each trial. (**H**) Example PCA projection of the moment-to-moment RNN activity (colormap indicates track position). (**I**) Total variance explained by the principal components for network-wide activity across maps (top three principal components, red points). (**J**) Normalized manifold misalignment scores across models (0, perfectly aligned; 1, p=0.25 of shuffle). (**K**) Cosine similarity between the latent state and position input and output weights onto the remap dimension (left) and the position subspace (right) (error bars, sem; N = 15 models).

The online version of this article includes the following figure supplement(s) for figure 1:

**Figure supplement 1.** Single recurrent neural network (RNN) units remap heterogeneously.

**Figure supplement 2.** Recurrent neural network (RNN) geometry is interpretable.

behavioral states). Thus, we developed a task that requires a single neural circuit to navigate a 1D circular environment while inferring transiently cued, discrete latent state changes. We hypothesized that these task constraints would produce a manifold structure similar to that observed in Low et al. (*Figure 1D*).

We trained RNNs with $N = 248$ units to integrate a 1D velocity input along a circular environment (equivalent to a 1D virtual track with seamless teleportation, as in *Low et al., 2021*; *Figure 1E*, top) and to concurrently remember a binary latent state signal (*Figure 1E*, bottom). Trained RNNs achieve high performance in both tasks, indicating that they can correctly form a persistent representation of each latent state while maintaining a stable position estimate across states (100% correct state estimation; average angular position error after 300 steps, mean ± standard deviation: 8.13° ± 0.51°; *Figure 1F*). To visualize trial-by-trial RNN activity, we provided the trained model with nonnegative velocity inputs and divided the resulting session into track traversals, labeling each traversal by the predominant latent state (*Figure 1G*, red, context 1; black, context 2). As in biological data (*Figure 1C*), single units and network-wide activity alternated between distinct maps of the same environment across the two latent states (*Figure 1G*). Similar to biological neurons, single RNN units remapped heterogeneously (*Figure 1—figure supplement 1*). Units changed their spatial firing field locations to a similar extent as biological neurons, but changes in firing rate were more common in the model units.

When we projected the hidden layer activity into the subspace defined by the first three principle components, the activity occupied two distinct rings, where position along each ring corresponded to position on the linear track (*Figure 1H*, red to blue color map). Together, these top three components explained ~50% of the variance (*Figure 1I*, red points). As in *Low et al., 2021*, we used Procrustes shape analysis to demonstrate that these rings were more aligned in high-dimensional activity space than expected by chance for all trained models, such that the position along one ring matched the position on the second ring (*Figure 1J*). Thus, these task constraints are sufficient to organize randomly initialized synaptic weights into a network model that qualitatively reproduces the remapping dynamics and representational geometry that *Low et al., 2021* observed in the entorhinal cortex.

## RNN geometry is interpretable

We next considered the network geometry in more detail, asking how much of the geometry arises necessarily from the task design. Some basic features of the network structure follow intuitively from the components of the task. First, the RNN must maintain an estimate of a 1D circular position, which is best achieved through approximate ring attractors (*Cueva et al., 2019*). Thus, we expect the model to form two ring attractors, one for each of the two distinct latent state conditions. Second, the network must track two statistically independent information streams and should, therefore, develop separate orthogonal subspaces for each stream (*Kaufman et al., 2014*). One subspace, the 'position subspace,' should contain the position tuning curves for all neurons, as well as the RNN readout weights for the position. The other subspace, the 'remapping dimension,' should be tuned to changes in the latent state and contain the readout weights for the state. We confirmed that these dimensions were orthogonal to one another (*Figure 1K*, *Figure 1—figure supplement 2*, Methods), such that changes in the latent state do not interfere with changes in position and vice versa.

However, as we show in detail below, the task does not require the two-ring manifolds to be strictly aligned in high-dimensional activity space. The RNN maintains a stable position estimate in spite of switches in the latent state—which we call 'remapping events'—that may occur anywhere along the circular track. To do so, the RNN must implement an invertible remapping function to match track locations across the two ring manifolds. This remapping function could be complex and high dimensional (*Figure 2A*), resulting in misaligned manifolds, or it could take the form of a simple translation in firing rate space (*Figure 2B*), resulting in aligned manifolds. The RNN could implement a complex remapping function using its recurrent dynamics, but this implementation could lead to a delay between the latent state signal and remapping. We therefore reasoned that the RNN might converge to the simpler configuration, allowing the linear input layer to implement remapping and thereby enabling rapid remapping in a single timestep.

It is useful to mathematically formalize these ideas to show that the alignment of the two ring manifolds is not strictly imposed by the task. Consider an RNN with $N$ neurons and, for simplicity, consider a discrete grid of $P$ position bins indexed by $p \in \{1, \ldots, P\}$. Let $x_p^{(1)}$ denote an $N$-dimensional vector

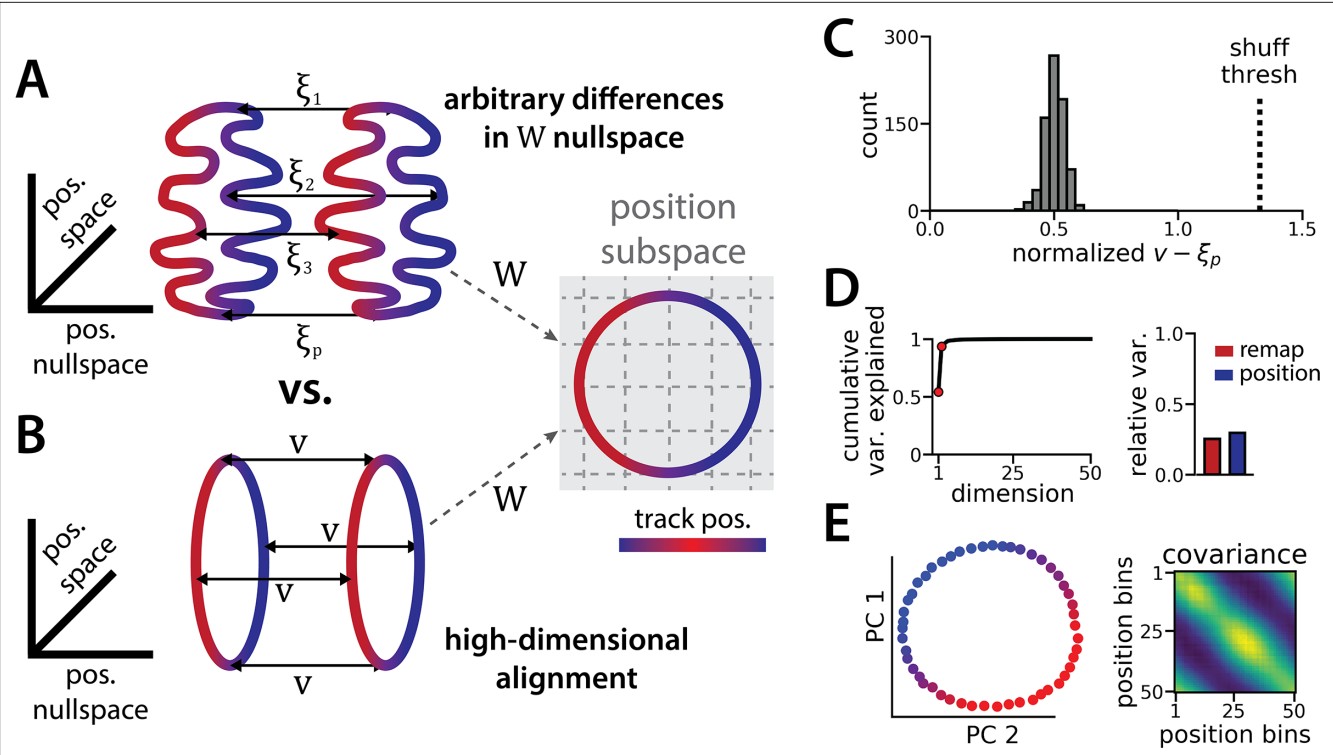

**Figure 2.** The ring attractors are more aligned than strictly required by the task. (**A**) Schematic illustrating how the two attractor rings could be misaligned in the nullspace of the position output weights (left) while still allowing linear position decoding (right) (color map, track position; solid arrows, remapping vectors; dashed arrow, projection onto the position subspace). (**B**) Schematic illustrating perfect manifold alignment (colors as in A) (**C**) Normalized difference between the true remapping vectors for all position bins and all models and the ideal remapping vector (dashed line, p=0.025 of shuffle; n=50 position bins, 15 models). (**D**) Dimensionality of the remapping vectors for an example model. (Right) Total variance explained by the principal components for the remapping vectors (red points, top two PCs). (Left) Relative variance is explained by the remapping vectors (red) and the position rings (blue) (1=total network variance). (**E**) The remapping vectors vary smoothly over the position. (Right) Projection of the remapping vectors onto the first two PCs. (Left) Normalized covariance of the remapping vectors for each position bin (blue, min; yellow, max).

corresponding to the neural firing rates in spatial bin $p$ along the first ring attractor (i.e. position $p$ in state 1). Likewise, let $x_p^{(2)}$ denote the corresponding firing rate vector in the second ring attractor (i.e. position $p$ in state 2). We can compute $x_p^{(1)}$ and $x_p^{(2)}$ by averaging the RNN activations across many simulated trials, similar to how spatial tuning curves are estimated in biological data.

Let $W$ denote the $2 \times N$ matrix holding the readout layer weights used to decode angular position $\theta$ by predicting $\cos\theta$ and $\sin\theta$ (see Methods). The linearity of this decoder imposes a constraint that $Wx_p^{(1)} = Wx_p^{(2)}$ for all $p$; otherwise, the decoded position will erroneously depend on which latent state is active. Importantly, this constraint does not imply that the two rings must have the same shape nor that they must be aligned. To see this let $\xi_1, \ldots, \xi_P$ denote any arbitrary set of $N$-dimensional vectors in the nullspace of $W$ (i.e. we have $W\xi_p = 0$) and define $x_p^{(2)} = x_p^{(1)} + \xi_p$. Then it is easy to see that the constraint is satisfied,

$$Wx_p^{(1)} = Wx_p^{(1)} + W\xi_p = W\left(x_p^{(1)} + \xi_p\right) = Wx_p^{(2)}$$

Because each $\xi_p$ was chosen arbitrarily from the nullspace of $W$, an $(N-2)$-dimensional subspace, there are many configurations of the two rings that are compatible with linear decoding of position (**Figure 2A**). An alternative, lower-dimensional network geometry would instead remap along a constant $N$-dimensional translation vector $v$, such that we have $x_p^{(2)} = x_p^{(1)} + v$ (approximately) for all positions $p$ (**Figure 2B**).

We now explore each of these intuitions in our data to see how well the trained RNN matches our expectations. First, how closely do the manifolds match the best alignment wherein $x_p^{(2)} = x_p^{(1)} + v$? We computed the empirical remapping vectors $\xi_p = x_p^{(2)} - x_p^{(1)}$ for each position bin and verified that $W\xi_p \approx 0$ for all positions (mean ± sem: $9\times10^{-6} \pm 10^{-5}$). We then defined $v$ to be the average of these remapping vectors, $v = \langle \xi_p \rangle$. If the manifolds were perfectly aligned then we would observe $\xi_p = v$ for all positions $p$.

We instead find that there is some variability in the remapping vectors, such that $\xi_1, \ldots, \xi_P$ are not exactly equal to one another (**Figure 2C**). Indeed, when we perform PCA on the $P \times N$ matrix formed by concatenating the vectors $\xi_1, \ldots, \xi_P$, we find that remapping dimensions lie within a 2-dimensional subspace (**Figure 2D and E**), in contrast to our original conjecture that remapping vectors would be effectively zero-dimensional (i.e. $\xi_p = v$ for all positions). Nonetheless, the idealized model in which each $\xi_p \approx v$ is a much better fit to the observed RNN dynamics than would be expected by chance. When we randomly rotate the orientation of the two rings in the nullspace of $W$, we find that this approximation is much worse (**Figure 2C**, dashed line).

Altogether, these findings suggest that RNN models trained on a simultaneous latent state inference and navigation task converge to a geometrically simple solution out of the space of all possible, high-dimensional solutions. This simpler solution recapitulates the geometry of entorhinal cortical dynamics during remapping in virtual reality environments (**Low et al., 2021**). Notably, neither the RNN nor the biological data are consistent with the simplest 3-dimensional solution, as evidenced by the imperfect ring alignment (**Low et al., 2021** and **Figure 1J**), the variable remapping vectors (**Figure 2C–D**), and the dimensionality of the network dynamics (which is >3; **Figure 1I**).

## RNN dynamics follow two-ring attractor manifolds

While neural manifold geometry can provide clues about the computational mechanisms at play in the system, one advantage of RNN models is that we can precisely characterize the structure and logic of their dynamics using tools from nonlinear systems analysis (**Maheswaranathan et al., 2019**; **Sussillo and Barak, 2013**). As we describe below, these tools reveal several insights into the underlying network computations that are not easy to experimentally demonstrate in biological networks.

Each RNN defines a nonlinear, discrete-time dynamical system, $x_{t+1} = f(x_t, u_t)$, where $f(\cdot, \cdot)$ is a nonlinear function parameterized by synaptic weight matrices and $u_1, \ldots, u_T$ defines a sequence of latent state cues and velocity inputs to the network. Using methods pioneered in **Sussillo and Barak, 2013**, we used numerical optimization to identify *approximate fixed points*, which are N-dimensional vectors $x_*$ that satisfy $f(x_*, u) \approx x_*$ for a specified input $u$. In particular, we studied the case where $u = 0$, corresponding to a situation where no velocity or context input is provided to the network. Intuitively, the network should approach a fixed point when no velocity or context input is provided because the position and latent state are unchanging.

The fixed points of the RNN provide a backbone for understanding its dynamics. While the global RNN dynamics are complex and nonlinear, the dynamics near any fixed point $x_*$ can be approximated as a linear dynamical system governed by the $N \times N$ Jacobian matrix of partial derivatives $\partial f_i / \partial x_j$ evaluated at $x = x_*$ and $u = 0$ (see Methods).

We computed these Jacobian matrices across 988 fixed points in the trained RNN shown in **Figure 1**. Roughly 60% of these fixed points were located on one of the two previously described ring manifolds and largely had marginally stable linear dynamics (largest Jacobian eigenvalue $\approx 1$; **Figure 3**, color-coded green). The remaining fixed points were located between the two ring manifolds and had unstable dynamics (largest Jacobian eigenvalue >1; **Figure 3**, color-coded gold). In essence, this analysis confirms that the RNN dynamics indeed implemented a pair of ring attractors. Furthermore, a collection of unstable fixed points form a boundary between the two stable ring attractor basins. In line with observations by **Sussillo and Barak, 2013** on a discrete flip-flop task, these intermediate fixed points are unstable along a small number of dimensions (i.e. saddle fixed points; **Figure 3D**, gold points) which 'funnel' neural activity to the appropriate location during a remapping event.

This interpretation is supported by examining the principal eigenvector—i.e., the eigenvector associated with the largest magnitude eigenvalue—for each fixed point. For the fixed points along the two ring attractors, this eigenvector corresponds to a slow dimension along which $x$ does not grow or decay (i.e. its associated eigenvalue $\lambda \approx 1$; **Figure 3D**, green points). Consistent with a mechanism for integrating positional information, these eigenvectors were nearly orthogonal to the remapping

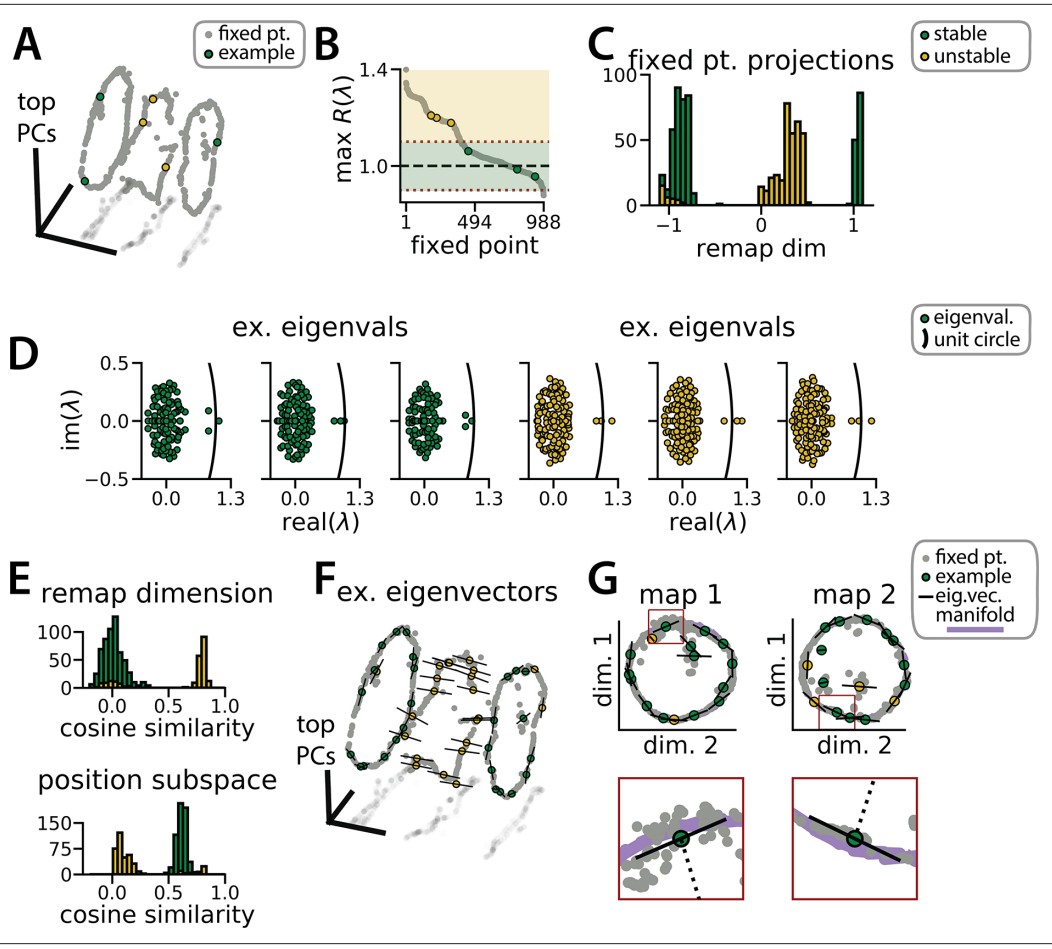

**Figure 3.** Recurrent neural network (RNN) dynamics follow stable ring attractor manifolds mediated by a ring of saddle points. (**A**) Fixed points (gray) reside on the two ring manifolds and in a ring between them (gray, all fixed points; colors indicate example fixed points shown in B and D). (**B**) The maximum real component of each fixed point (shading, marginally stable or unstable points; colored points, examples from A). Dashed red lines indicate cut-off values for fixed points to be considered marginally stable ($\lambda \approx 1$). (**C**) Projection of fixed points onto the remapping dimension (−1, map 1 centroid; 1, map 2 centroid; 0, midpoint between manifolds). (**D**) Distribution of eigenvalues (colored points) in the complex plane for each example fixed point from (**A**) (black line, unit circle). (**E**) Cosine similarity between the eigenvectors associated with the largest magnitude eigenvalue of each fixed point and the remap dimension (top) or the position subspace (bottom). (**F**) Eigenvector directions for 48 example fixed points (black lines, eigenvectors). (**G**) Projection of the example fixed points closest to each manifold onto the respective position subspace (top) and zoom on an example fixed point for each manifold (bottom, red box) (purple, estimated activity manifold; dashed line, approximate position estimate). Green indicates marginally stable points and gold indicates unstable points throughout.

dimension and aligned with the position subspace (*Figure 3E–G*, green). Conversely, for the unstable fixed points, the principal eigenvector corresponds to a dimension along which $x$ moves rapidly away from the fixed point (i.e. its associated eigenvalue $\lambda > 1$; *Figure 3D*, gold points). Consistent with a mechanism for 'funneling' activity during remapping events, these eigenvectors were aligned with the remapping dimension and nearly orthogonal to the position subspace (*Figure 3E–F*, yellow).

## Aligned toroidal manifolds emerge in a 2D generalization of the task

Virtual 1D tracks are an ideal setting to experimentally study spontaneous remapping: the environmental cues can be tightly controlled and it is possible to sample an identical spatial trajectory hundreds of times, such that remapping events can be easily identified from the neural activity alone. But navigation is often studied in 2D environments, in which it is more difficult to control the animal's experience and the animal can pursue an essentially infinite number of trajectories through the

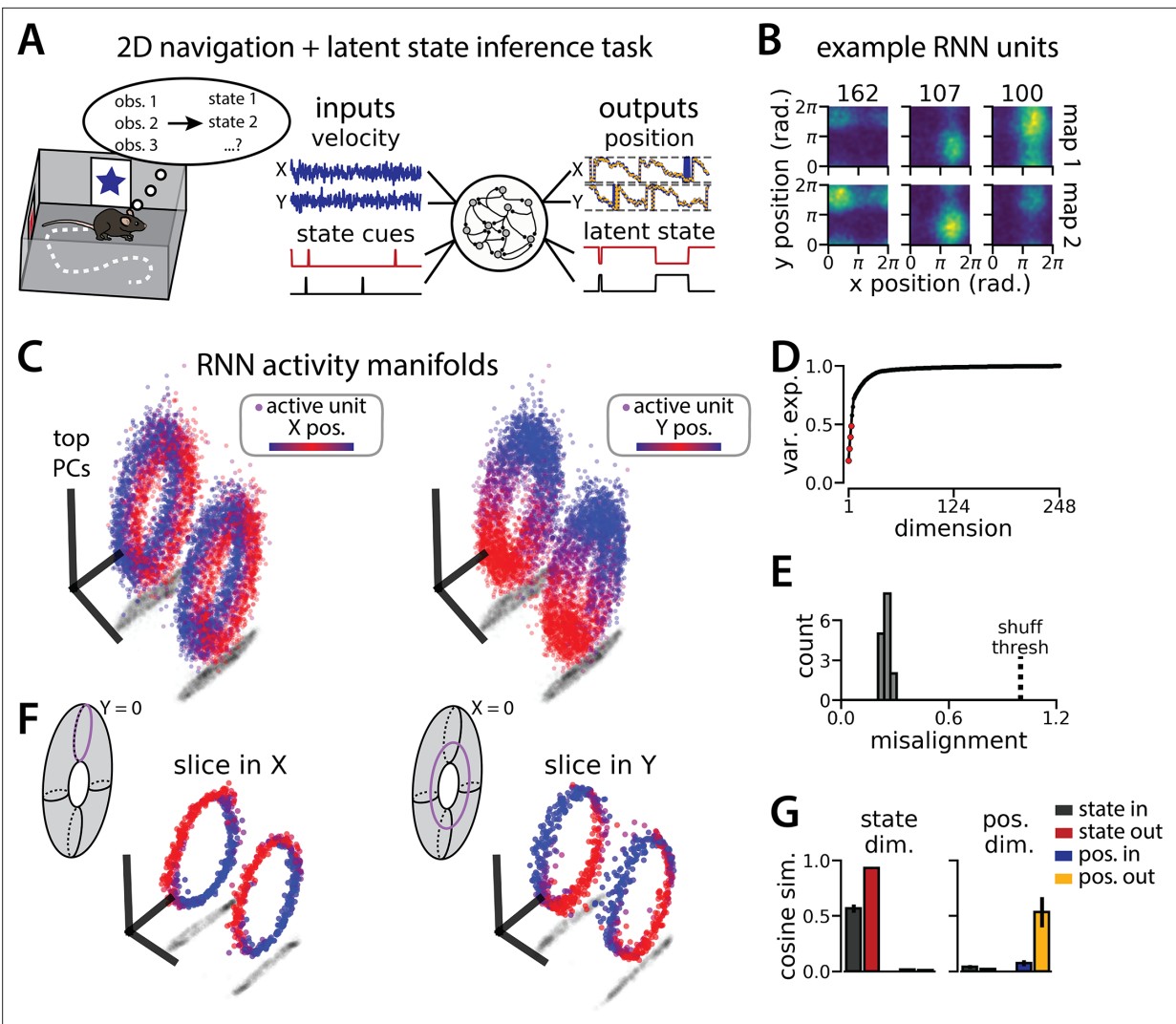

**Figure 4.** An recurrent neural network (RNN) model of 2D navigation and context inference remaps between aligned toroidal manifolds. (**A**) (Left) Schematic illustrating the 2D navigation with simultaneous latent state inference task. (Right) As in *Figure 1E*, but RNN models were trained to integrate two velocity inputs (X, Y) and output a 2D position estimate, in addition to simultaneous latent state inference. (**B**) Position-binned activity for three example units in latent state 1 (top) and latent state 2 (bottom)(colormap indicates normalized firing rate; blue, minimum; yellow, maximum). (**C**) Example principal components analysis (PCA) projection of the moment-to-moment RNN activity from a single session into three dimensions (colormap indicates position; left, X position; right, Y position). Note that the true tori are not linearly embeddable in 3 dimensions, so this projection is an approximation of the true torus structure. (**D**) Average cumulative variance explained by the principal components for network-wide activity across maps (top four principal components, red points). (**E**) Normalized manifold misalignment scores for all models (0, perfectly aligned; 1, p=0.25 of shuffle). (**F**) Example PCA projection of slices from the toroidal manifold where Y (left) or X (right) position is held constant, illustrating the substructure of RNN activity. (**G**) Cosine similarity between the latent state and position input and output weights onto the remap dimension (left) and the position subspace (right), defined for each pair of maps (error bars, sem; n=15 pairs, 15 models).

environment. Thus, while it is of interest to understand what remapping in the entorhinal cortex might look like in 2D spaces, it remains challenging to identify spontaneous remapping in biological data. In contrast, the RNN modeling framework that we have developed here can be readily generalized to 2D spatial environments. Are the computational solutions identified by the RNNs fundamentally different in this case? Or do RNNs use similar geometric structures and algorithmic principles across these related tasks?

To investigate this question, we again trained models to simultaneously integrate velocity inputs and estimate latent state from transient state cues, but this time we provided two velocity inputs and asked the models to estimate position on a 2D circular track (*Figure 4A*, right). As before, the models performed well on both components of the task (mean loss ± sem: position estimate, 0.036 ± 1.1 ×

$10^{-3}$; latent state estimate, $0.002 \pm 1.9 \times 10^{-5}$; n = 15 models) (**Figure 4A**), and single unit activity was modulated by both spatial position and latent state (**Figure 4B**). When we projected the activity into a subspace defined by the first three principal components, the activity occupied two distinct toroidal manifolds with the position on each torus corresponding to the position in the 2D space (**Figure 4C**). Notably, each toroidal manifold alone is reminiscent of networks trained to store two circular variables without remapping (**Cueva et al., 2021**). In keeping with these qualitative observations, four principal components explained ~50% of the variance in network activity (**Figure 4D**), and the manifolds were again highly, though not perfectly, aligned in the full-dimensional activity space (**Figure 4E**). By holding either the horizontal (X) or vertical (Y) position variable constant during the RNN simulation, we recover a pair of 1D-aligned ring manifolds (**Figure 4F**). That is, we can recover the geometry of the original 1D task (see **Figure 1**) by taking 'slices' through the toroidal manifolds. The remapping and position dimensions were again orthogonalized in these models (**Figure 4G**). Thus models trained on a 2D navigation task with latent state inference identified a geometrically similar solution to those trained on a 1D task. These findings demonstrate that spontaneous remapping is possible in 2D and may operate under similar mechanisms as in 1D.

## Manifold alignment generalizes to three or more maps

It is simplest to consider remapping as switches between two maps, but neural activity can conceivably switch between any number of maps. Indeed, while Low et al. most commonly observed remapping between two maps of the same virtual track, they occasionally found transitions between more than two maps (**Low et al., 2021**). Because these 'multi-map' sessions were rare, Low et al. predominantly limited their analysis to the more common '2-map' sessions. Nonetheless, there were notable similarities between the 2-map and multi-map sessions. In particular, remapping was correlated with changes in running speed and position was preserved across remapping events (**Low et al., 2021**). We reasoned that we could study the geometry of 'multi-map' sessions using RNN models to gain insight into what we might expect to see in biological data. In particular, do multiple ring manifolds corresponding to multiple spatial maps emerge and are these ring manifolds still geometrically aligned with each other?

We trained models to integrate a 1D velocity input, while tracking three (instead of two) binary state switch cues (**Figure 5A**). Models performed well on both task components (mean loss ± sem: position estimate, $0.013 \pm 3.6 \times 10^{-4}$; latent state estimate, $0.0039 \pm 4.9 \times 10^{-5}$; n=15 models) and single unit activity was modulated by both spatial position and latent state (**Figure 5B**). Importantly, we found that the same essential geometry of the original task was preserved. When we visualized each pair of maps using PCA, the ring manifolds were again qualitatively aligned (**Figure 5C**, right). Projecting the activity from all three maps into the same subspace revealed that they were further organized as vertices of an equilateral triangle (**Figure 5C**, left)—i.e., the acute angle between any two remapping dimensions was 60° (**Figure 5D**). Again, the network geometry was relatively low-dimensional (4 principal components explained ~60% of the variance; 14 principal components, ~90% of the variance) (**Figure 5E**). Procrustes analysis revealed that all pairs of manifolds were highly aligned relative to chance, with a similar degree of alignment across ring manifold pairs and across RNNs (**Figure 5F**). Finally, positional and latent state information were orthogonalized, as before (**Figure 5G**).

In **Figure 5—figure supplement 1**, we show that RNNs are capable of solving this task with larger numbers of latent states (more than three; for simplicity, we consider up to 10 states). Furthermore, the RNN dynamics and geometry generalize accordingly: each latent state is associated with a different ring attractor and every pair of ring attractors is highly aligned. Motivated by these observations, we revisited a subset of experimental sessions from **Low et al., 2021** (N=4 sessions from 2 mice) that exhibited remapping between 3–4 stable maps of the same virtual track (**Figure 6A, B**; **Figure 6—figure supplement 1**) for a pilot comparison with the RNN models, which we hope will inspire future experimental analysis. As Low et al., we first confirmed that these remapping events did not reflect recording probe movement by comparing the waveforms from different maps across the session, which were highly stable (**Figure 6B**, right; **Figure 6—figure supplement 1**).

To examine the manifold structure of the neural data, we projected the neural tuning curves associated with each pair of maps into the subspace defined by the first three principle components. In many sessions, population-wide activity across the two maps occupied distinct, qualitatively aligned rings, where the position along each ring corresponded to the position along the virtual track

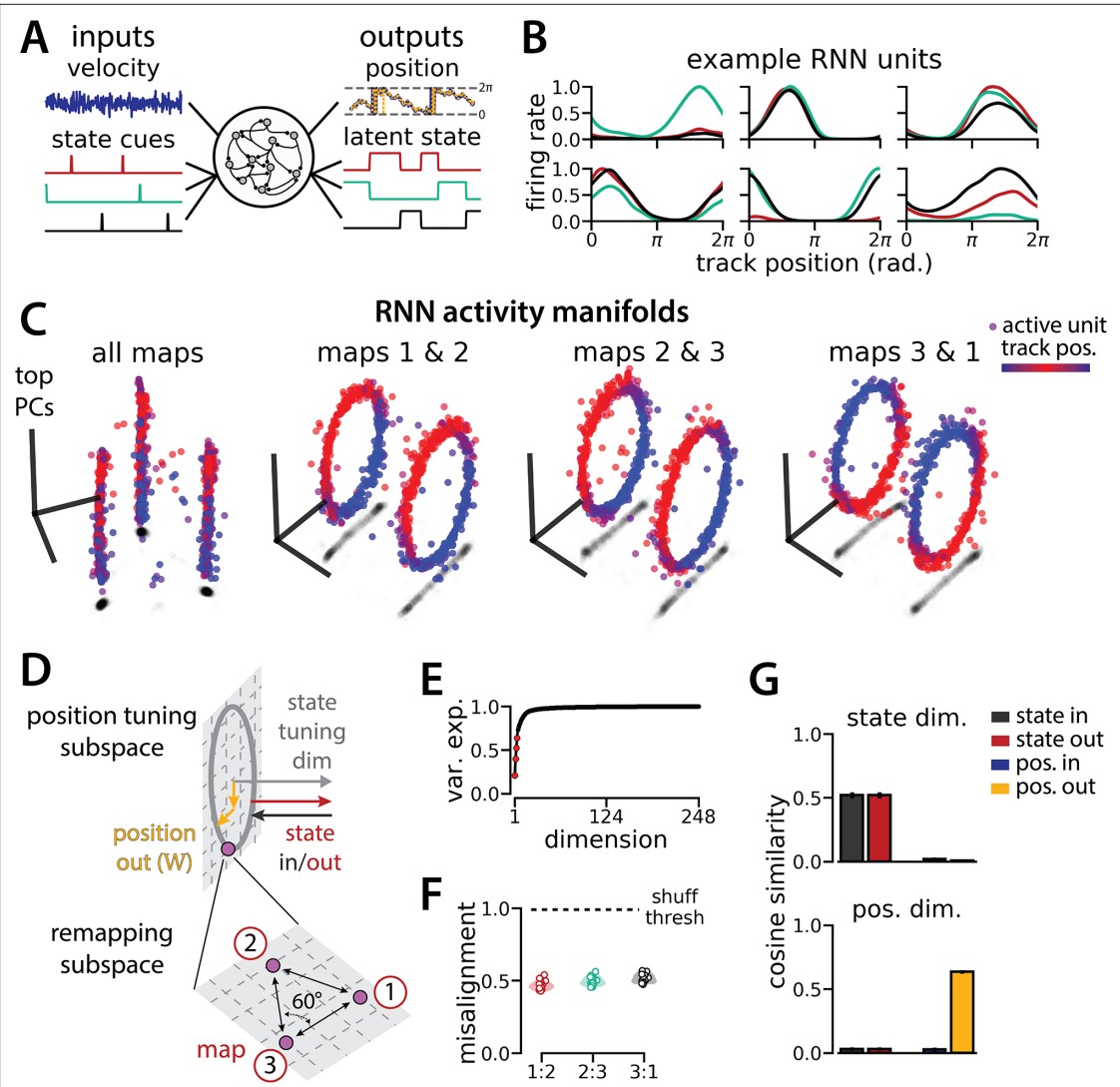

**Figure 5.** An recurrent neural network (RNN) model of 1D navigation and identification of three latent states remaps between three aligned ring manifolds. (**A**) (Left) RNN models were trained to navigate in 1D (velocity signal, top) and discriminate between three distinct contexts (transient, binary latent state signal, bottom). (Right) Example showing high prediction performance for the position (top) and context (bottom). (**B**) Position-binned activity for six examples single RNN units, split by context (colors as in (**A**)). (**C**) Example principal components analysis (PCA) projection of the moment-to-moment RNN activity into three dimensions (colormap indicates track position) for all three contexts (left) and for each pair of contexts (right). (**D**) Schematic: (Top) The hypothesized orthogonalization of the position and context input and output weights. (Bottom) Across maps, corresponding locations on the 1D track occupy a 2D remapping subspace in which the remapping dimensions between each pair are maximally separated (60°). (**E**) Total variance explained by the principal components for network-wide activity across maps (top four principal components, red points). (**F**) Normalized manifold misalignment scores between each pair of maps across all models (0, perfectly aligned; 1, p=0.25 of shuffle). (**G**) Cosine similarity between the latent state and position input and output weights onto the remap dimension (left) and the position subspace (right), defined for each pair of maps (error bars, sem; n=45 pairs, 15 models).

The online version of this article includes the following figure supplement(s) for figure 5:

**Figure supplement 1.** Manifold geometry generalizes for up to 10 latent states.

(*Figure 6C*). Procrustes analysis revealed that these pairs of maps were largely more aligned than expected by chance (*Figure 6D*; 13/18 map pairs more aligned than shuffle). Notably, five map pairs from one mouse (three in session A, two in session D) were not aligned (*Figure 6D*, black and teal points), suggesting that manifold alignment does not always emerge in biological data. Finally, we asked whether the remapping dimensions from the biological sessions were organized symmetrically, as in the model (*Figure 6E*, left). We found that there was a range of acute angles between pairs of

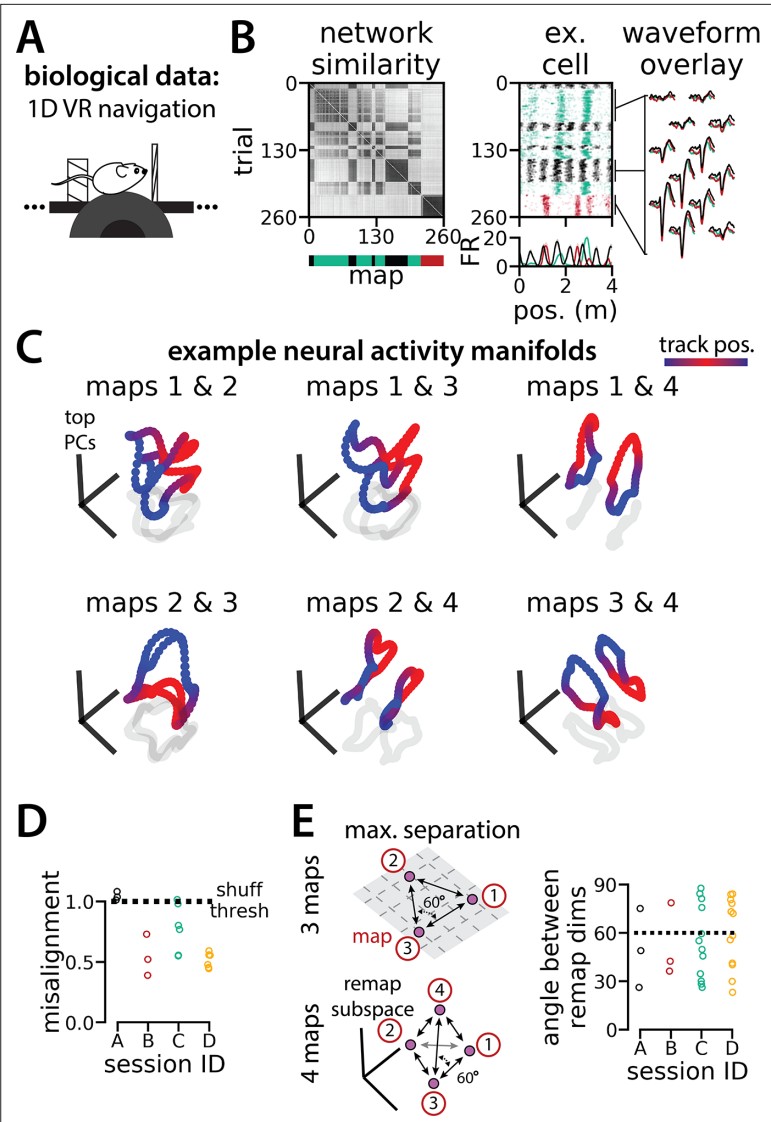

**Figure 6.** Biological recordings with more than two maps recapitulate many geometric features of the recurrent neural network models. (**A - B**) are modified from *Low et al., 2021*. (**A**) Schematic: Mice navigated virtual 1D circular-linear tracks with unchanging environmental cues and task conditions. Neuropixels recording probes were inserted during navigation. (**B**) Examples from the 3-map Session A. (Left/top) Network-wide trial-by-trial correlations for the spatial firing pattern of all co-recorded neurons in the same example session (color bar indicates correlation). (Left/bottom) k-means map assignments. (Middle) An example medial entorhinal cortex neuron switches between three maps of the same track (top, raster; bottom, average firing rate by position; teal, map 1; red, map 2; black, map 3). (Right) Overlay of average waveforms sampled from each of the three maps. (**C**) Principal components analysis (PCA) projection of the manifolds associated with each pair of maps from the 4-map Session D (color bar indicates virtual track position). (**D**) Normalized manifold misalignment scores between each pair of maps across all sessions (0, perfectly aligned; 1, p=0.25 of shuffle). (**E**) (Left) Schematic: maximal separation between all remapping dimensions for 3 and 4 maps. (Right) Angle between adjacent pairs of remapping dimensions for all sessions (dashes, ideal angle).

The online version of this article includes the following figure supplement(s) for figure 6:

**Figure supplement 1.** Medial entorhinal cortex can remap between 3 or 4 maps and many geometric features are preserved.

remapping dimensions (*Figure 6E*, right), suggesting that there was more asymmetry in the biological network geometry than in the model.

## Discussion

Previous experimental studies have found that neurons in the medial entorhinal cortex change their firing patterns in response to changes in task conditions, behavioral state, or visual and motor cues (*Bant et al., 2020*; *Boccara et al., 2019*; *Butler et al., 2019*; *Campbell et al., 2021*; *Campbell et al., 2018*; *Hardcastle et al., 2017b*). In virtual reality environments, these remapping events can recruit neurons across the entorhinal cortical circuit to rapidly switch between distinct maps of the same track (*Campbell et al., 2021*; *Low et al., 2021*). Here, we used RNN models to explore a normative hypothesis that these remapping dynamics reflect hidden state inference (*Colgin et al., 2008*; *Sanders et al., 2020*). We showed that RNNs initialized from random synaptic weights recapitulate the essential features of biological data—aligned ring manifolds (*Low et al., 2021*)—when trained to remember a binary latent state variable and to simultaneously integrate a velocity signal in a circular environment. RNNs learn to represent positional and state change information in orthogonal subspaces such that navigation and latent state inference co-occur without interference. Furthermore, we demonstrated that the geometry and algorithmic principles of this solution readily generalize to more complex tasks including navigation in 2D environments and tasks involving three or more latent states. These findings provide a jumping-off point for new analyses of remapping in neural data, which we demonstrated in a pilot analysis of neural data from *Low et al., 2021*.

These results complement an existing body of theoretical and experimental work on the neural basis of navigation. *Fenton et al., 2010* proposed that the hippocampus constructs multiple spatial maps that are anchored to different landmarks; when the animal's attention switches between these reference points, hippocampal cells remap (see also *Kubie et al., 2020*). This proposal is consistent with the idea that the hippocampal circuit groups navigational episodes into discrete categories by combining internal context with external landmarks (*Colgin et al., 2008*; *Fuhs and Touretzky, 2007*; *Sanders et al., 2020*). Related experimental work demonstrates that an animal's prior experience with an environment can shape how the hippocampus delineates these categories (*Plitt and Giocomo, 2021*). Each of these hypotheses can be seen as layering a discrete latent variable (e.g. changes in reference landmarks, task context, or prior experience) on top of a distributed neural code of position, which are the essential ingredients of our RNN task. While we draw explicit comparisons with spontaneous remapping in the entorhinal cortex (*Low et al., 2021*), *Sheintuch et al., 2020* reported similar experimental findings in the hippocampus, highlighting the broad relevance of this remapping phenomenon and our modeling efforts. We explored these topics in a general modeling framework applicable to any circuit that supports navigation through physical space and even navigation of abstract cognitive spaces (*Aronov et al., 2017*; *Constantinescu et al., 2016*; *Whittington et al., 2020*).

In our task, discrete switches in the latent state are signaled by brief impulses to the RNN, such that the navigational circuit must maintain a persistent representation of the latent state based on these transient cues. This simple task design allowed us to clearly identify the minimal set of constraints that produces aligned ring attractors. In particular, our results suggest that aligned ring attractors could emerge even if upstream circuits trigger latent state changes and signal these changes to downstream navigational circuits. Indeed, Low et al. found that remapping was correlated with brief decreases in running speed (*Low et al., 2021*), suggesting that this temporary behavioral state change—which is known to have widespread impacts on global brain activity (*Stringer et al., 2019*)—may serve as an external trigger of remapping in the entorhinal cortex. Extensions to our task could build on this basic framework by asking the network to infer state changes given a noisy input or using a more complex interaction with the environment (e.g. through reinforcement learning paradigms (*Uria et al., 2020*).

The mechanisms of remapping in biological circuits are still poorly understood, but have been modeled using multistable attractor dynamics for several decades (*Samsonovich and McNaughton, 1997*). Classically, these models were engineered and hand-tuned to produce the desired attractor dynamics. In contrast, RNN models are indirectly engineered by specifying task constraints and a learning algorithm (*Yang and Wang, 2021*). Thus, our observation that trained RNNs produce multistable attractor manifolds is nontrivial because different solutions might have, in principle, emerged. Despite this key similarity, there are notable differences between our models and classical

multistable attractor models. Classical models typically store completely decorrelated spatial maps (*Samsonovich and McNaughton, 1997*), while our RNNs produce distinct maps that are, by construction, perfectly correlated in the position readout dimensions. *Romani and Tsodyks, 2010* studied the effects of adding correlation to spatial maps in forward-engineered multistable attractor networks, as did *Low et al., 2021*. Fundamentally, these and other forward-engineered models provide insights into *how* neural circuits may remap, but do not answer *why* they do so. We investigated the latter question in this work by identifying a minimal set of task constraints that provide a putative explanation for why the entorhinal cortex spontaneously remaps.

Other work has studied remapping in trained artificial networks performing navigation (*Schøyen et al., 2022*; *Uria et al., 2020*). Unlike our results, these papers typically consider remapping across different physical environments. *Whittington et al., 2020* propose a normative model and a neural circuit that supports non-spatial remapping, which is perhaps most similar to the task constraints we studied. However, our investigation focused on a simpler and more targeted computational task to draw a tighter link to a specific biological finding and to perform a deeper examination of the resulting population geometry and dynamical structure.

While we were motivated to study remapping in the specific context of navigational circuits, our results have broader implications for understanding how RNNs perform complex, context-dependent computations. This topic has attracted significant interest. For example, RNNs trained in many computational tasks develop modular neural populations and dynamical motifs that are re-used across tasks (*Driscoll et al., 2022*; *Yang et al., 2019*). When RNN architecture is explicitly designed to include dedicated neural subpopulations, these subpopulations can improve model performance on particular types of tasks (*Beiran et al., 2021*; *Dubreuil et al., 2022*). Thus, there is an emerging conclusion that RNNs use simple dynamical motifs as building blocks for more general and complex computations, which our results support. In particular, aligned ring attractors are a recurring, dynamical motif in our results, appearing first in a simple task setting (two maps of a 1D environment) and subsequently as a component of RNN dynamics in more complex settings (e.g. as sub-manifolds of toroidal attractors in a 2D environment, see *Figure 4*). We can, therefore, conceptualize a pair of aligned ring manifolds as a dynamical 'building block' that RNNs utilize to solve higher-dimensional generalizations of the task. Intriguingly, our novel analysis of neural data from *Low et al., 2021* revealed that similar principles may hold in biological circuits—when three or more spatial maps were present in a recording, the pairs of ring manifolds tended to be aligned.

Ultimately, our model provides a strong foundation for future experimental investigations of the functional role of remapping in navigational circuits. Our findings suggest that latent state changes can drive remapping; an experimental task that explicitly requires animals to report a latent internal state would provide substantial insight into this hypothesis. We also identify concrete predictions for how the representational geometry of neural populations generalizes from the dynamics found in 1D virtual reality environments (*Campbell et al., 2021*; *Low et al., 2021*) to more complex settings. We found direct support for one of these predictions by re-analyzing an existing experimental dataset. Our work, therefore, provides a parsimonious, plausible, and testable model for the neural population geometry of remapping navigational circuits under a variety of task conditions.

## Materials and methods

**Key resources table**

| Reagent type (species) or resource | Designation | Source or reference | Identifiers | Additional information |
|---|---|---|---|---|
| Software, algorithm | SciPy ecosystem of open-source Python libraries | *Harris et al., 2020*; *Hunter, 2007*; *Jones et al., 2001* | https://www.scipy.org/ | libraries include numpy, matplotlib, scipy, etc. |
| Software, algorithm | scikit-learn | *Pedregosa et al., 2012* | https://pytorch.org/docs/stable/nn.html | |
| Software, algorithm | PyTorch | *Paszke et al., 2019* | https://pytorch.org/docs/stable/nn.html | |

## Resource availability

### Lead contact
Further information and requests for resources and reagents should be directed to and will be fulfilled by the Lead Contact, Alex H. Williams (alex.h.williams@nyu.edu).

### Materials availability
This study did not generate new unique reagents.

## Experimental model and subject details

### RNN model and training procedure
We examined Elman RNNs ('vanilla' RNNs), which are perhaps the simplest RNN architecture capable of theoretically representing any nonlinear dynamical system (**Hammer, 2000**) and which can be viewed as an approximation to continuous time firing rate models of neural circuits (**Song et al., 2016**). At each time index $t \in \{1, \cdots, T\}$ the activation vector of $N$ hidden units is denoted by $x_t \in R^N$. Loosely, we can think of $x_t$ as the firing rates of $N$ neurons in a biological circuit at time $t$. The activation vector is updated according to:

$$x_{t+1} = ReLU\left(Ax_t + Bu_t + \beta\right)$$

where $ReLU\left(x\right)$ denotes a rectifying linear unit function (i.e. an element-wise maximum between the vector $x$ and a vector of zeros), $A \in R^{N \times N}$ is a matrix holding the recurrent synaptic connection weights, $u_t \in R^M$ is a vector of input signals at time $t$, $B \in R^{N \times M}$ is a matrix holding the input connection weights, and $\beta \in R^N$ is a vector holding bias terms for each hidden unit. The output of the network at time $t$ is defined by:

$$y_t = Cx_t + \alpha$$

where $y_t \in R^L$ is a vector of $L$ output units, $C \in R^{L \times N}$ is a matrix holding output connection weights, and $\alpha \in R^L$ is a vector holding bias terms for each output unit. Finally, the initial condition $x_0 \in R^N$ for each dynamical sequence was set by:

$$x_0 = Dz + \gamma$$

where $z \in R^{M_0}$ is a vector of $M_0$ inputs used to define the initial condition, $D \in R^{N \times M_0}$ is a matrix holding connection weights, and $\gamma \in R^{M_0}$ is a vector holding bias terms. The connection weights were randomly initialized from the uniform distribution over $\left(-\sqrt{\frac{1}{N}}, \sqrt{\frac{1}{N}}\right)$, which is the default initialization scheme in PyTorch. As described below, the vector $z$ is used to define the initial position on the circular track, which is randomized in each trial. Altogether, these equations define an RNN model with trainable parameters $\{A, B, \beta, C, \alpha, D, \gamma\}$.

The number of inputs, $M$, and outputs, $L$, varied depending on the computational task the RNN was trained to perform. Specifically, $M$ is given by the number of latent states ('contexts') plus the number of spatial dimensions. Thus, for the 1D navigation task with binary state cues diagrammed in **Figure 1E**, the number of network inputs was $M = 3$ (whereas the tasks diagrammed in **Figure 4A** and **Figure 5A** each have $M = 4$ inputs). The number of network outputs, $L$, is given by the number of latent states plus two times the number of spatial dimensions. Thus, for the task diagrammed in **Figure 1E**, $L = 4$ (for **Figure 4A**, $L = 6$; for **Figure 5A**, $L = 5$). The additional spatial output dimensions can be understood as follows: Due to the periodic boundary conditions, the network must output a predicted spatial position $\theta_t \in [0, 2\pi)$ for each spatial dimension. Predicting this raw angular position would require the network to implement something akin to an $\arctan\left(\cdot\right)$ function. Because this function is highly nonlinear and discontinuous, the linear readout layer of the RNN will struggle to predict $\theta_t$ directly in a fashion that is numerically stable. We therefore trained the networks to predict $\sin\theta_t$ and $\cos\theta_t$ for each spatial dimension, which requires an extra factor of two spatial output dimensions. Similarly, for the initial condition $x_0$, the number of input variables $M_0$ is given by two times the number of spatial dimensions, and the input vector $z$ is formed by concatenating $\sin\theta_0$ and $\cos\theta_0$ for each spatial dimension.

The network was trained by randomly generated input sequences with a ground truth target output. The input vector at each time step, $u_t$, contained the angular velocity along each spatial dimension as well as state change cues (see schematic in *Figure 1E*). The output vector at each time step, $y_t$, contains disjoint dimensions that predict the spatial position and the latent state or 'context' (see schematic in *Figure 1F*). For each sequence, the overall loss function is a sum of two terms: (*i*) the mean-squared-error between the ground truth sine and cosine of angular position, $\sin\theta_t$ and $\cos\theta_t$, and the network's prediction of these terms, and (*ii*) the cross-entropy of the true latent state and the network's prediction (see **torch.nn.CrossEntropyLoss** class in the PyTorch library *Paszke et al., 2019*).

We trained networks with $N = 248$ hidden units using stochastic gradient descent with a batch size of 124 sequences and gradient clipping (gradient norm clipped to be less than or equal to 2). At the beginning of training, we trained RNNs on sequence lengths of $T = 1$ and increased the sequence length by one every 50 parameter updates. We performed 30,000 parameter updates, so that by the end of training the RNNs were training on sequence lengths of $T = 600$. We found that this gradual increase in task complexity along with gradient clipping was necessary to achieve good performance. Intuitively, training on short sequences at the beginning helps the network learn suitable parameter values for $\{C, \alpha, D, \gamma\}$ without worrying about the typical challenges (e.g. exploding and vanishing gradients) associated with RNN training. Then, the remaining parameters $\{A, B, \beta\}$ can be fine-tuned with gradually increasing sequence lengths.

Each sequence was randomly generated. For each spatial dimension, the initial angular position, $\theta_0$, was sampled uniformly between $[0, 2\pi]$. The angular velocity at each time step was given by $\Delta\theta_t = \underline{\theta} + \epsilon_t$ where $\underline{\theta}$ denotes the mean velocity and $\epsilon_t$ was sampled randomly from a normal distribution with a mean of zero and a standard deviation of 0.3 radians. For each sequence the mean velocity, $\underline{\theta}$, was sampled from a normal distribution with a mean of zero and a standard deviation of 0.1 radians. The initial latent state was chosen randomly from the available states. State change cues occurred randomly according to a homogeneous Poisson process with an expected rate of one state change per 50-time steps. In *Figure 5*, we trained networks to switch between three or more states—for each state change one of the inactive states was chosen uniformly at random to be the new active state. State changes were cued by a pulse lasting two-time steps.

For *Figure 1G*, the trained model was provided velocity inputs with an initial position of $\theta_0 = 0$ and non-negative angular velocity at each time step, $\Delta\theta_t = |\underline{\theta} + \epsilon_t|$, so that the RNN output would follow a trial structure comparable to the biological data. Similarly, state changes occurred less frequently (at an expected rate of once per 500-time steps) to better match the biologically observed remapping rate. For comparison with the biological data, we truncated each sequence to remove incomplete track traversals and concatenated 50 sequences into a single session. For visualization purposes, we computed the smoothed, position-binned (n bins = 50) firing rates for 5 example units and labeled each track traversal according to the most commonly reported latent state for that traversal.

## Mice

All experimental data reported here were collected for a previous publication by *Low et al., 2021*, and were approved by the Institutional Animal Care and Use Committee at Stanford University School of Medicine. More information on data collection and analyses can be found in Method Details, below, and in the Methods section of *Low et al., 2021*.

## Method details

### Manifold geometry analyses

We used Procrustes shape analysis (*Gower and Dijksterhuis, 2004*) according to the methods described by *Low et al., 2021* to determine the degree to which manifolds from different maps were aligned in the high-dimensional activity space. Briefly, we divided the track into 50 position bins and computed the average activity for all units within each position bin for each latent state to obtain an estimate of the manifold associated with each map. We then mean-centered these manifolds and rescaled them to have unit norms. We compute the root-mean-squared error (RMSE) between these two manifolds (the 'observed' RMSE). We then find the rotation matrix that optimally aligns the two manifolds and calculate the RMSE between the optimally aligned manifolds. We report the observed RMSE relative to the RMSE after optimal (misalignment = 0) and random (misalignment =

1) rotation. For *Figure 5* and *Figure 5—figure supplement 1*, which had more than two latent states, we computed this score for all pairs of manifolds.

In *Figures 1 and 3–5* we consider the network in terms of subspaces tuned to the two task components—position and latent state. We define the 'position subspace' as a two-dimensional subspace containing the position-binned average firing rates of all units. We divided the track into 250 position bins and computed the average activity for all units within each position bin for each latent state. To find the position subspace across maps—as in *Figure 1K*, *Figure 3E*, *Figure 4G*, and *Figure 5G* — we performed a 2-factor Principal Components Analysis (PCA) on the position-binned activity across both latent states. To find the position subspace for a single map—as in *Figure 3G* —we performed 2-factor PCA on the average activity from just one latent state. We define the 'remapping dimension' as the dimension separating the manifold centroids, which we find by computing the average activity for each unit within each map and taking the difference across the two maps.

In *Figures 1K, 4G and 5G*, and *Figure 1—figure supplement 2B*, we calculate the angles between the input and output weights and the position subspace or remapping dimension. To find this angle, we calculated the cosine similarity between each weight vector and each subspace. Cosine similarity of 0 indicates that the weights were orthogonal to the subspace, while a similarity of 1 indicates that the weight vector was contained within the subspace.

## Fixed point analysis

We numerically identified fixed points according to the methods described in *Sussillo and Barak, 2013* Briefly, we used stochastic gradient descent to minimize $\left\| x - ReLU\left(Ax + \beta\right) \right\|_2$ over hidden layer activation vectors $x$. Values of $x$ that minimize the expression close to zero, correspond to approximate fixed points of the recurrent RNN dynamics when the input is held constant $u_t = 0$. At each numerical fixed point $x_*$, we use standard autodifferentiation tools in PyTorch to compute the $N \times N$ Jacobian matrix $\frac{\partial x_{t+1}}{\partial x_t}$ evaluated at $x = x_*$ . The eigenvalues and eigenvectors of this matrix then provide a local linear dynamical approximation to the full system as explained in *Sussillo and Barak, 2013* and in the main text.

## Single unit analysis

To characterize single unit remapping properties for *Figure 1—figure supplement 1*, we performed the rate remapping versus global remapping analysis described in *Low et al., 2021* For each model, we computed the average firing rate for all units in each map, smoothing with a Gaussian filter (standard deviation, two position bins). We then calculated the percent change in peak firing rate (i.e. rate remapping). To compute a spatial dissimilarity score (i.e. global remapping), we subtracted 1 from the cosine similarity between firing rate vectors (a dissimilarity score of 0 indicates identical spatial firing, and 1 indicates orthogonal spatial representations).

## Experimental data

The experimental data included in *Figure 1*, *Figure 6*, and *Figure 6—figure supplement 1* were collected for a previous publication (*Low et al., 2021*). Briefly, mice were trained to navigate a 1D virtual reality track with tower landmarks and floor cues to provide optic flow. The landmarks repeated seamlessly every 400 cm such that the track was circular-linear. The mice received randomly distributed, visually cued rewards within the middle 300 cm of the track. During behavior, neural activity was recorded using Neuropixels 1.0 silicon recording probes (*Jun et al., 2017*), which were acutely inserted into the medial entorhinal cortex. Behavioral and neural data were processed as described by *Low et al., 2021*; *Figure 1A–D* are modified from Low et al.

The pilot analyses in *Figure 6* and *Figure 6—figure supplement 1* are performed on a subset of the data from Low et al. (n=684 cells from four sessions in two mice) (*Low et al., 2021*). As described in that publication, we used k-means clustering to divide these sessions into 3 or 4 maps. We then assessed the trial-by-trial spatial stability of the population-wide neural activity within each map in order to restrict our analysis to stable trials. We divided the session according to the k-means map labels and computed the Pearson correlation between the position-binned firing rates (n bins = 80) of all neurons for each pair of trials within each map. We excluded trials that were spatially unstable from our analysis (average correlation with all other trials <0.25). (We performed the same analysis of trial-by-trial spatial stability to obtain the similarity matrices in *Figure 1C and G*).

To assess the geometry of the neural population activity, we used the k-means cluster centroids as an estimate for the neural activity manifold associated with each map. We then performed Procrustes shape analysis to assess manifold alignment and identified the remapping dimensions, as described above.

To ensure that remapping was not an artifact of probe movement or multi-unit activity, we compared the spike waveforms for all cells across remapping events, as described in *Low et al., 2021* Briefly, we identified the longest epoch of trials for each map and extracted waveforms for 100 spikes for each cell from each epoch. We then computed the average waveforms within each epoch. To determine waveform similarity, we computed the Pearson correlation between the vectorized average waveforms for each pair of maps and then calculated the average correlation across pairs. For all waveform analyses, we used waveforms from the 20 channels closest to the Kilosort2-identified depth for each cell.

*Figure 1A–D*, *Figure 1—figure supplement 1B*, and *Figure 6A–B* are modified from the following figure panels in *Low et al., 2021*: the graphical abstract (left/middle panel—schematic of entorhinal neural activity), Figure 1A, Figure 4B, Figure 7E (left), Figure S1H (middle right), and Figure S5A (left) and B (top). These figures were originally published under an Elsevier user license. The copyright holder has granted permission to publish under a CC BY 4.0 license.

## Quantification and statistical analysis

### Statistics

All data were analyzed in Python, using the scipy stats library to compute statistics. Unless otherwise noted, all tests are two-sided, correlation coefficients represent Pearson's correlation, and values are presented as mean ± standard error of the mean (SEM). Statistical tests are listed following the relevant result given in the Results, figure legend, or Method Details. Unless otherwise stated, $p < 0.05$ was taken as the criterion for significance.

## Acknowledgements

We thank Dmitriy Aronov and Selmaan Chettih for providing feedback on the manuscript. We thank Scott Linderman, members of the Aronov Lab, members of the Giocomo Lab, and members of the Williams Lab for discussions and feedback. This work was supported by funding from the Wu Tsai Neurosciences Institute under Stanford Interdisciplinary Graduate Fellowships (to IICL); the Office of Naval Research (N00141812690), the Simons Foundation (SCGB 542987SPI), NIMH (1R01MH126904-01A1 and U19NS118284), the Vallee Foundation, and the James S McDonnell Foundation (to LMG); and the Simons Foundation (to AHW).

## Additional information

### Funding

| Funder | Grant reference number | Author |
|---|---|---|
| Wu Tsai Neurosciences Institute, Stanford University | Stanford Interdisciplinary Graduate Fellowship | Isabel IC Low |
| Office of Naval Research | N00141812690 | Lisa M Giocomo |
| Simons Foundation | SCGB 542987SPI | Lisa M Giocomo |
| National Institute of Mental Health | 1R01MH126904-01A1 | Lisa M Giocomo |
| National Institute of Mental Health | U19NS118284 | Lisa M Giocomo |
| Vallee Foundation | | Lisa M Giocomo |
| James S. McDonnell Foundation | | Lisa M Giocomo |

| Funder | Grant reference number | Author |
|---|---|---|
| Simons Foundation | | Alex H Williams |

The funders had no role in study design, data collection and interpretation, or the decision to submit the work for publication.

## Author contributions

Isabel IC Low, Conceptualization, Formal analysis, Investigation, Visualization, Methodology, Writing – original draft, Writing – review and editing; Lisa M Giocomo, Conceptualization, Supervision, Funding acquisition, Writing – review and editing; Alex H Williams, Conceptualization, Formal analysis, Supervision, Funding acquisition, Investigation, Visualization, Methodology, Writing – original draft, Writing – review and editing

## Author ORCIDs

Isabel IC Low (iD) http://orcid.org/0000-0001-6465-8459
Lisa M Giocomo (iD) http://orcid.org/0000-0003-0416-2528
Alex H Williams (iD) http://orcid.org/0000-0001-5853-103X

Reviewer #1 (Public Review): https://doi.org/10.7554/eLife.86943.3.sa1
Reviewer #2 (Public Review): https://doi.org/10.7554/eLife.86943.3.sa2
Reviewer #3 (Public Review): https://doi.org/10.7554/eLife.86943.3.sa3
Author Response: https://doi.org/10.7554/eLife.86943.3.sa4

# Additional files

## Supplementary files

• MDAR checklist

## Data availability

No new data were generated for this manuscript, as it is a computational study. Code to train the RNN models and reproduce the figures of the paper are provided in a GitHub repository (copy archived at *Williams and Low, 2023*).

The following previously published dataset was used:

| Author(s) | Year | Dataset title | Dataset URL | Database and Identifier |
|---|---|---|---|---|
| Giocomo LM, Low IC | 2022 | Low et al. (2021) Dynamic and reversible remapping of network representations in an unchanging environment. Neuron | https://data.mendeley.com/datasets/hntn6m2pgk/1 | Mendeley Data, 10.17632/hntn6m2pgk.1 |

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
