## [Editor Report · eLife assessment]

This **important** work provides evidence that artificial recurrent neural networks can be used to investigate neural mechanisms underlying reversible remapping of spatial representations. Authors perform **convincing** state of the art analyses showing how population activity preserves the encoding of spatial position despite remappings due to the tracking of an internal variable. This paper will be of interest to neuroscientists studying contextual computations, neural representation of space and links between artificial neural networks and the brain.

---

## [Referee Report · Reviewer #1 (Public Review)]

Based on a recent report of spontaneous and reversible remapping of spatial representations in the enthorhinal cortex (Low et al 2021), this study sets out to examine possible mechanisms by which a network can simultaneously represent a positional variable and an uncorrelated binary internal state. To this end, the authors analyse the geometry of activity in recurrent neural networks trained to simultaneously encode an estimate of position in a one-dimensional track and a transiently-cued binary variable. They find that network activity is organised along two separate ring manifolds. The key result is that these two manifolds are significantly more aligned than expected by chance, as previously found in neural recordings. Importantly, the authors show that this is not a direct consequence of the design of the model, and clarify scenarios by which weaker alignment could be achieved. The model is then extended to a two-dimensional track, and to more than two internal variables. The latter case is compared with experimental data that had not been previously analysed.

Strengths:

rigorous and careful analysis of activity in trained recurrent neural networksparticular care is taken to show that the obtained results are not a necessary consequence of the design of the modelthe writing is very clear and pleasant to readclose comparison with experimental dataextensions beyond the situations studied in experiments (two-dimensional track, more than two internal states)

Weaknesses:

no major weaknesses(minor) the comparison with previous models of remapping could be expanded

Altogether the conclusions claimed by the authors seem to be strongly supported and convincing.

---

## [Referee Report · Reviewer #2 (Public Review)]

This important work presents an example of a contextual computation in a navigation task through a comparison of task driven RNNs and mouse neuronal data. Authors perform convincing state of the art analyses demonstrating compositional computation with valuable properties for shared and distinct readouts. This work will be of interest to those studying contextual computation and navigation in biological and artificial systems.

This work advances intuitions about recent remapping results. Authors trained RNNs to output spatial position and context given velocity and 1-bit flip-flops. Both of these tasks have been trained separately, but this is the first time to my knowledge that one network was trained to output both context and spatial position. This work is also somewhat similar to previous work where RNNs were trained to perform a contextual variation on the Ready-Set-Go with various input configurations (Remington et al. 2018). Additionally findings in the context of recent motor and brain machine interface tasks are consistent with these findings (Marino et al in prep). In all cases contextual input shifts neural dynamics linearly in state space. This shift results in a compositional organization where spatial position can be consistently decoded across contexts. This organization allows for generalization in new contexts. These findings in conjunction with the present study make a consistent argument that remapping events are the result of some input (contextual or otherwise) that moves the neural state along the remapping dimension.

The strength of this paper is that it tightly links theoretical insights with experimental data, demonstrating the value of running simulations in artificial systems for interpreting emergent properties of biological neuronal networks. For those familiar with RNNs and previous work in this area, these findings may not significantly advance intuitions beyond those developed in previous work. It's still valuable to see this implementation and satisfying demonstration of state of the art methods. The analysis of fixed points in these networks should provide a model for how to reverse engineer and mechanistically understand computation in RNNs.

I'm curious how the results might change or look the same if the network doesn't need to output context information. One prediction might be that the two rings would collapse resulting in completely overlapping maps in either context. I think this has interesting implications about the outputs of the biological system. What information should be maintained for potential readout and what information should be discarded? This is relevant for considering the number of maps in the network. Additionally, I could imagine the authors might reproduce their current findings in another interesting scenario: Train a network on the spatial navigation task without a context output. Fix the weights. Then provide a new contextual input for the network. I'm curious whether the geometric organization would be similar in this case. This would be an interesting scenario because it would show that any random input could translate the ring attractor that maintains spatial position information without degradation. It might not work, but it could be interesting to try!

I was curious and interested in the authors choice to not use activity or weight regularization in their networks. My expectation is that regularization might smooth the ring attractor to remove coding irrelevant fluctuations in neural activity. This might make Supplementary Figure 1 look more similar across model and biological remapping events (Line 74). I think this might also change the way authors describe potential complex and high dimensional remapping events described in Figure 2A.

Overall this is a nice demonstration of state-of-the-art methods to reverse engineer artificial systems to develop insights about biological systems. This work brings together concepts for various tasks and model organisms to provide a satisfying analysis of this remapping data.

---

## [Referee Report · Reviewer #3 (Public Review)]

This important work provides convincing evidence that artificial recurrent neural networks can be used to model neural activity during remapping events while an animal is moving along a one-dimensional circular track. This will be of interest to neuroscientists studying the neural dynamics of navigation and memory, as well as the community of researchers seeking to make links between artificial neural networks and the brain.

Low et al. trained artificial recurrent neural networks (RNNs) to keep track of their location during a navigation task and then compared the activity of these model neurons to the firing rates of real neurons recorded while mice performed a similar task. This study shows that a simple set of ingredients, namely, keeping track of spatial location along a one-dimensional circular track, along with storing the memory of a binary variable (representing which of the two spatial maps are currently being used), are enough to obtain model firing rates that reproduce features of real neural recordings during remapping events. This offers both a normative explanation for these neural activity patterns as well as a potential biological implementation.

One advantage of this modeling approach using RNNs is that this gives the authors a complete set of firing rates that can be used to solve the task. This makes analyzing these RNNs easier, and opens the door for analyses that are not always practical with limited neural data. The authors leverage this to study the stable and unstable fixed points of the model. However, in this paper there appear to be a few places where analyses that were performed on the RNNs were not performed on the neural data, missing out on an opportunity to appreciate the similarity, or identify differences and pose challenges for future modeling efforts. For example, in the neural data, what is the distribution of the differences between the true remapping vectors for all position bins and the average remapping vector? What is the dimensionality of the remapping vectors? Do the remapping vectors vary smoothly over position? Do the results based on neural data look similar to the results shown for the RNN models (Figures 2C-E)?

There are many choices that must be made when simulating RNNs and there is a growing awareness that these choices can influence the kinds of solutions RNNs develop. For example, how are the parameters of the RNN initialized? How long is the RNN trained on the task? Are the firing rates encouraged to be small or smoothly varying during training? For the most part these choices are not explored in this paper so I would interpret the authors' results as highlighting a single slice of the solution space while keeping in mind that other potential RNN solutions may exist. For example, the authors note that the RNN and biological data do not appear to solve the 1D navigation and remapping task with the simplest 3-dimensional solution. However, it seems likely that an RNN could also be trained such that it only encodes the task relevant dynamics of this 3-dimensional solution, by training longer or with some regularization on the firing rates. Similarly, a higher-dimensional RNN solution may also be possible and this would likely be necessary to explain the more variable manifold misalignment reported in the experimental data of Low et al. 2021 as opposed to the more tightly aligned distribution for the RNNs in this paper. However, thanks to the modeling work done in this paper, the door has now been opened to these and many other interesting research directions.

---

## [Author Response]

The following is the authors' response to the original reviews.

**Reviewer #1 (Recommendations For The Authors):**
This is a list of suggestions the authors could use to improve the details of the manuscript:- it is not immediately clear what is meant by "modular" on line 38 and the corresponding paragraph. This aspect is not mentioned or developed in the Results.- the discussion of remapping vectors on lines 119-137 is particularly illuminating. It could have been interesting to generate surrogate manifolds separated by arbitrary remapping vectors and see how much the alignment metric (Procrustes shape) is sensitive to the dimensionality or amplitude of remapping vectors.- A visual comparison between Fig 1 D and H suggests a difference between the manifold geometry in experiments and in the model. It seems that the embedding dimensionality of ring manifolds is higher in the data than in the model. Is that the case? It could have been interesting to explore how much embedding dimensionality influences the alignment metric.- I could not find information about the initialization of the connectivity weights. An important possibility is that the degree of alignment (and the organization of remapping vectors) depends on the strength of initial random connectivity.- It might have been interesting to comment on the relationship between the top three PCS in Fig1 and the three readout vectors. To which extent are they aligned?- I found panels C and G in Fig 1 somewhat difficult to read. In panel C, the remapping seems to be aligned to the same position across all trials. This is not the case in panel G. I am not certain what the comparison is meant to convey, but it would help to have a similar alignment in C and G. Similarly, I was not sure what to conclude from the matrix in the right part of panel C, perhaps the legend should be expanded.- the comparison with remapping models of Misha Tsodyks could be expanded. The current discussion implies that the model of Romani & Tsodyks leads to less alignment than found in trained networks, but no direct evidence is given for that statement as far as I can tell.
**Reviewer #2 (Recommendations For The Authors):**
Minor points:All mentions of 'modularity' should be replaced with 'compositionality'.I found Supplementary Figure 2 highly confusing. I thought it was meant to help understand the analysis in Figure 1K and related figures. In the end, I never really understood what was happening in these figures. Do authors make perturbations along these different coding dimensions and compare the resulting maps? I wasn't sure what exactly the authors were calculating cosine similarity for. Maybe more exposition on this in the methods would help other readers as well.Was there any behavioral difference when the maps were not aligned?Why did the authors only go up to 10 contexts? Was this dependent on size of the network? Sorry if I missed this.Are remapping event aligned to unit axes? Would this change with different nonlinearities? This could be interesting in the context of (Driscoll et all 2022) and (Wittington et al 2022).
**Reviewer #3 (Recommendations For The Authors):**
Cueva, Ardalan, et al. 2021 arXiv:2111.01275 showed that RNNs trained to remember two circular variables develop a toroidal geometry to store this information, so consider citing this in your section on the toroidal manifolds.

We thank the reviewers for their thoughtful comments. We appreciate that all three reviewers affirmed the importance of our work and the rigor of our approach. We believe that no major weaknesses were identified by the reviews. In our view, the comparisons between recurrent neural network models and experimental data are one of the most important contributions of our work, and all reviewers agreed that this was a core strength of the manuscript.

The reviewers highlighted several future modeling directions that are raised by our results and that we did not explore in the manuscript. For example, Reviewer 2 suggests that we train networks on a navigation task alone, freeze the weights, and then train on a context discrimination task. We agree that this kind of contextual learning paradigm is of interest and could provide insight into biological remapping, such as that observed by Low et al. (2021). We also agree with Reviewer 3’s broader point that “There are many choices that must be made when simulating RNNs and there is a growing awareness that these choices can influence the kinds of solutions RNNs develop.” It is notable that we were able to reproduce the qualitative features of the experimental data without finely tuning hyperparameters (we used default settings in PyTorch layers), using a very basic training protocol (gradient descent with gradient clipping), and without adding any hand crafted regularization (though we agree that regularization could make the RNN solution look even more like the data).

We believe that readers will benefit from reading the reviewers' suggestions, which are insightful and well-motivated. Having weighed the reviewer comments carefully, we feel that our manuscript stands as a complete scientific story. We hope that the public reviewer comments will inspire future investigations to fully explore these possibilities and unpack their outcomes at a level of detail that would not be possible in the context of our manuscript.

Thus, we have chosen to implement the following minor changes suggested by the reviewers, which we hope will improve the clarity of the text and figures (summarized below). These changes do not alter the fundamental content of the manuscript.

Text:

We corrected a few minor typos.We updated the citations to follow the eLife citation style.To address comments from Reviewers 1 and 2: we reworded the final paragraph of the Introduction (p. 3) to remove the term “modularity” and clarify our main finding. Those sentences now read, “The RNN geometry and algorithmic principles readily generalized from a simple task to more complex settings. Furthermore, we performed a new analysis of experimental data published in Low et al.26 and found a similar geometric structure in neural activity from a subset of sessions with more than two stable spatial maps.”To address comments from Reviewer 1: in the first paragraph of the Results section *A recurrent neural network model of 1D navigation and context inference remaps between aligned ring manifolds* (p. 3), we added the sentence, “Remapping was not aligned to particular track positions, rewards, or landmarks.” to clarify that experimental result from Low et al. (2021).To address comments from Reviewer 3: in the final paragraph of the Results section *Aligned toroidal manifolds emerge in a 2D generalization of the task* (p. 11) we clarified that models were trained “to estimate position on a 2D circular track.” We also added a citation to Cueva, Ardalan et al. (2021) with the following sentence, “Notably, each toroidal manifold alone is reminiscent of networks trained to store two circular variables without remapping.”

To address a question from Reviewer 2: in the final paragraph of the Results section *Manifold alignment generalizes to three or more maps* (p. 13), we added the following clarification: “In Supplemental Figure 3, we show that RNNs are capable of solving this task with larger numbers of latent states (more than three**;** for simplicity, we consider up to 10 states).”To address a comment from Reviewer 1: in the fourth paragraph of the Discussion (p. 17), we removed the sentence, “Notably, our model captured aspects of the data that these previous forward-engineered models did not explore—namely, that the ring manifolds corresponding to the correlated spatial maps were much more aligned than expected by chance and than strictly required by the task.” to focus on the key point in the following sentence that, “forward-engineered models provide insights into *how* neural circuits may remap, but do not answer *why* they do so.”To address comments from Reviewers 1 and 2: we reworded the penultimate paragraph of the Discussion (p. 17–18) to clarify our findings and remove the term “modularity” (except when referencing papers that themselves use that term (Driscoll et al., 2022; Yang et al., 2019)). Those sentences now read:

“When RNN architecture is explicitly designed to include dedicated neural subpopulations, these subpopulations can improve model performance on particular types of tasks (Beiran et al., 2021; Dubreuil et al., 2022). Thus, there is an emerging conclusion that RNNs use simple dynamical motifs as building blocks for more general and complex computations, which our results support. In particular, aligned ring attractors are a recurring, dynamical motif in our results, appearing first in a simple task setting (2 maps of a 1D environment) and subsequently as a component of RNN dynamics in more complex settings (e.g., as sub-manifolds of toroidal attractors in a 2D environment, see Figure 4). We can therefore conceptualize a pair of aligned ring manifolds as a dynamical “building block” that RNNs utilize to solve higher-dimensional generalizations of the task. Intriguingly, our novel analysis of neural data from Low et al. (2021) revealed that similar principles may hold in biological circuits—when three or more spatial maps were present in a recording, the pairs of ring manifolds tended to be aligned.”

To address questions from Reviewers 2 and 3: in the first paragraph of the Methods section *RNN Model and Training Procedure* (p. 21), we added the sentence: “The connection weights were randomly initialized from the uniform distribution *U*(−√1/N, √1/N), which is the default initialization scheme in PyTorch.”

To address a question from Reviewer 2: we added a third paragraph to the Methods section *Manifold Geometry Analysis* (p. 23), as follows:

“In Figure 1K, 4G, 5G, and Supplementary Figure 2B, we calculate the angles between the input and output weights and the position subspace or remapping dimension. To find this angle, we calculated the cosine similarity between each weight vector and each subspace. Cosine similarity of 0 indicates that the weights were orthogonal to the subspace, while a similarity of 1 indicates that the weight vector was contained within the subspace.”

To address a question from Reviewer 1: we added the following sentence to the second paragraph of the Methods section *Experimental Data* (p. 24), “We performed the same analysis of trial-by-trial spatial stability to obtain the similarity matrices in Figure 1C and G.”

Figures and legends:

To address a question from Reviewer 1: in Figure 1C and G, we added x-axis labels to the similarity matrices to clarify that these are trial-by-trial correlations.To address a question from Reviewer 1: we expanded the Figure 1C legend to clarify the experimental results as follows:

*Old legend:*

(C, left) An example medial entorhinal cortex neuron switches between two maps of the same track (top, raster; bottom, average firing rate by position; red, map 1; black, map 2). (C, right/top) Network-wide trial-by-trial correlations for the spatial firing pattern of all co-recorded neurons in the same example session (colorbar indicates correlation). (C, right/bottom) k-means map assignment.

*New legend:*

(C, left) An example medial entorhinal cortex neuron switches between two maps of the same track (top, spikes by trial and track position; bottom, average firing rate by position across trials from each map; red, map 1; black, map 2). (C, right/top) Correlation between the spatial firing patterns of all co-recorded neurons for each pair of trials in the same example session (dark gray, high correlation; light gray, low correlation). The population-wide activity is alternating between two stable maps across blocks of trials. (C, right/bottom) K-means clustering of spatial firing patterns results in a map assignment for each trial.

To address comments from Reviewer 3: in the legend of Figure 4C, we added the sentence “Note that the true tori are not linearly embeddable in 3 dimensions, so this projection is an approximation of the true torus structure.”To address a question from Reviewer 2: we expanded the legend for Supplementary Figure 2 to clarify the purpose of the figure schematics as follows:

*Old legend:*

(A) Schematic showing the orthogonalization of the position and context input and output weights.

(B) Reproduced from Figure 1K.

(C-D) Schematic: How a single velocity input (blue arrows) updates the position estimate (yellow to red points) from the starting position (blue points).

(C) Velocity input lies in the position tuning subspace (gray plane)(hypothetical). Note that the same velocity input results in different final positions.

(D) Velocity input is orthogonal to the position tuning subspace (observed).

(E) Schematic of possible flow fields in each of the three planes (numbers correspond to planes in C and D), which would result in the correct positional estimate given orthogonal velocity inputs at different positions (D).

*New legend:*

(A) Schematic showing the relative orientation of the position output weights and the context input and output weights to the position and state tuning subspaces.

(B) Reproduced from Figure 1K.

(C-D) Schematic to interpret why the position input weights are orthogonal to the position tuning subspace. These schematics illustrate how a single velocity input (blue arrows) updates the position estimate (yellow to red points) from a given starting position (blue points).

(C, not observed) Velocity input lies in the position tuning subspace (gray plane). Note that the same velocity input pushes the network clockwise or counterclockwise along the ring depending on the circular position

(D, observed) Velocity input is orthogonal to the position tuning subspace and pushes neural activity out of the subspace.

(E) Schematic of possible flow fields in each of three planes (numbers correspond to planes in C and D). We conjecture that these dynamics would enable a given orthogonal velocity input to nonlinearly update the position estimate, resulting in the correct translation around the ring regardless of starting position (as in D).